# Effect of Superabsorbent Polymer on the Mechanical Performance and Microstructure of Concrete

**DOI:** 10.3390/ma14123232

**Published:** 2021-06-11

**Authors:** Xiaoguo Zheng, Meng Han, Lulu Liu

**Affiliations:** 1Road and Bridge Department, Zhejiang Institution of Communications, Hangzhou 311112, China; zhengxiaoguo@zjvtit.edu.cn; 2State Key Laboratory of Coastal and Offshore Engineering, School of Civil Engineering, Dalian University of Technology, Dalian 116024, China; 2017221038@chd.edu.cn; 3Institute of Geotechnical Engineering, Southeast University, Nanjing 211189, China

**Keywords:** superabsorbent polymer (SAP), concrete, mechanical properties, volume stability, durability, microstructure

## Abstract

The internally cured material known as superabsorbent polymer (SAP) is an important innovation in concrete engineering technology. This paper investigates the effect of adding a polymer with superabsorbent capabilities on the physical and mechanical performance of concrete. The microstructure of the new hybrid concrete was also studied, and the influence of the polymer particle size and volume on the mechanical durability was evaluated. The mechanical properties of the new hybrid concrete, such as compressive strength, flexural strength, elastic modulus, and splitting tensile strength, were measured through laboratory experiments. The microstructure characteristics of the concrete were also investigated by scanning electron microscopy (SEM). The results show that shrinkage was reduced, while the volume stability of the concrete improved. Moreover, we found that cracking was reduced, while issues such as chloride penetration and freeze-thaw resistance were also improved. In addition, the SAP could effectively improve the microstructure of the concrete and refine the pore structure, as seen in the microscopic test. This paper helps to promote the development of internally cured material and improve technology for the prevention of concrete construction cracks.

## 1. Introduction

It is essential for sustainable development in concrete engineering to develop a resource-saving strategy. Improving the construction quality and working performance has been the core of the development of concrete engineering construction [1]. The working performance and service performances of concrete projects and materials are affected when they are in a coastal corrosive environment or an area of severe cold [2]. Furthermore, time consumption, wastage of resources and unsatisfactory curing effects are disadvantages to the externally cured method of concrete. Thus, internally cured technology for concrete is gradually being developed [3,4]. SAP, an internally cured material, is a macromolecular substance with a super water absorption capacity and strong water retention ability, which can effectively improve volume stability of the concrete, such as micro-cracks, autogenous shrinkage, and the drying shrinkage of concrete. The volume stability and durability of the concrete are also improved due to its strong water retention capacity and chemical hydrophilic group structure [5,6,7].

It has been suggested and demonstrated that the internally cured method can effectively improve the hydration process of concrete [7,8]. Research on the internal curing of concrete conducted by domestic and foreign scholars has mainly encompassed four aspects: (1) research on the mechanism of the internal curing of concrete; (2) the design method and preparation process of internal-curing materials for concrete; (3) the design of the mix ratio for the internal curing of concrete; and (4) research on the concrete performance improved by internally cured method [8,9]. The internal curing effect of SAP has primarily been evaluated in two parts. Firstly, the influence of SAP on the physical and mechanical properties, volume stability, durability, and microstructure of concrete, has been evaluated. Secondly, the interaction between the internal curing of SAP and the steel fiber has primarily been evaluated. This paper mainly studies the impact of the SAP particle size and content on the workability and microstructure of concrete in different mixture proportions [5,10].

Extensive research on the application of SAP materials in concrete engineering has been conducted by scholars. The effects of different kinds of fiber, internal curing agents, and extra water equivalent on concrete were discussed by [5,7]. The influence of SAP on the working performance, mechanical performances, volume stability, and durability of concrete under a low water‒cement ratio was analyzed [11,12,13]. On this basis, the creep characteristics of SAP in concrete were studied. The impact of the SAP and curing age on the pore distribution of concrete was also studied by using 3D volume analysis [14]. Furthermore, a study of the evolution law and the effect of time-varying damage by different internal curing agents on the macro and micro properties of concrete was discussed. The evolution law and prediction model of the concrete with SAP were also studied by [15,16,17]. The effect of SAP on the transition zone of the steel fiber interface in concrete and related properties was discussed by [9,10]. It has been ascertained from the above research results that the effect of SAP on concrete strength was controversial [13,18], and the frost resistance and durability of concrete can be effectively improved by SAP [19,20,21]. The autogenous shrinkage and dry shrinkage of concrete were reduced, and the tensile creep of the concrete could also be improved by 50% [15,16,22]. In addition, scanning electron microscopy (SEM), nuclear magnetic resonance (NMR), mercury intrusion porosimetry (MIP), and thermogravimetric analyzer (TGA) methods were adopted to discern the effect of SAP on the microstructure of concrete [14,23,24]. The influence mechanism of SAP on concrete performance was analyzed from the perspective of mineral composition, pore structure, and hydrated product morphology. The influence of the pore parameters of the concrete on the antifreeze performance was also studied [25,26,27]. An increase in the number of cement holes and small pores was obtained, which refined the concrete structure [28]. It was explained that the frost resistance and durability of concrete can be improved by SAP from the micro point of view [14,23,24]. The relationship between the SAP and the w/c ratio of concrete has a certain influence on the mechanical properties, volume stability, and durability of concrete mixed with SAP.

However, the influence of the SAP particle size, content, w/c ratio, and other factors on the mechanical properties and microstructure of concrete is vague, which needs to be determined further. Existing studies have not proposed prediction models and formulas for the SAP particle size, content, and mechanical properties under different water‒cement ratios. In addition, few quantitative studies on the degree of improvement by SAP for the dry shrinkage and durability of concrete have been conducted. The influence of SAP on the tensile strength of concrete also needs to be studied further. The range and mode of influence of the micropore structure characteristics of concrete with SAP on the compressive strength have not been determined, and need to be explored further. In addition, related literature on the importance of graded porosity is sparse. The influence mechanism of different pore classifications on the concrete strength has not been investigated in depth, and further research is needed.

Therefore, the influence of the SAP in different particle size and content on the mechanical performances, volume stability, and durability of concrete under different water‒cement ratios are studied, based on laboratory experiments. The prediction equations for the SAP particle size, content and mechanical performances are given. The influence mechanism of SAP on the properties of concrete is studied from three aspects: porosity, the average radius of pores, and the average spacing coefficient of pores. This can be concluded by using SEM, combined with optical microscopic image analysis. The micropore structure characteristics of concrete and their impact on the concrete strength are analyzed. The research results provide a theoretical basis and technical support for the application of SAP in engineering. They will also lay a foundation for research on the microscopic pore structure characteristics of concrete mixed with SAP.

## 2. Materials and Methods

### 2.1. Materials

SAP, a superfine white powder with strong and fast water absorption, was selected for this experiment. The SAP particles (0.15 mm, 0.3 mm, and 0.6 mm) used in the present study are sol-gel polymerized and irregular crushed powders. The chemistry of the 0.15 mm and 0.3 mm SAP was mainly low cross-linking acrylic acid-based sodium acrylic copolymer (Trusted Chemical Co. Ltd., Hangzhou, China), while the main chemical composition of the 0.6 mm SAP was polyacrylamide of the DS series (obtained from a Shanghai company). The tea-bag method [29,30] was adopted in this paper to calculate the water absorption capacity. Specific performance indicators are shown in Table 1. In addition, an ordinary Portland cement P.O. 42.5 was adopted and its initial and final setting time was 200 min and 275 min, respectively, and the volume stability was qualified. The specific performance indicators of SAP and cement are shown in Table 1 [3]. The river sand was used for fine aggregate and the coarse aggregate was selected from 5‒25mm crushed stone. The content of the crushed stone and its mud content was 16% and 0.2%, respectively. The specific materials used in the test are presented in Figure 1.

### 2.2. Test Procedure

To investigate the impact of SAP on the concrete performance, the effects of the SAP content, particle size, and different water‒cement ratios on the mechanical properties, volume stability, and durability of the concrete were studied. The crack prevention technology for concrete was improved, and a strong theoretical basis for the application of SAP was provided.

SAP with particle sizes of 0.6 mm, 0.3 mm, and 0.15 mm was used as raw materials. The amount of SAP was 0.2%, 0.4% and 0.6% of cement, respectively. The water‒cement ratios of 0.3 and 0.37 were used for comparison, and a total of 20 control experiments were designed. The specific mixing proportions were chosen based on actual bridge engineering, and also selected according to “the specification for the mix proportion design of ordinary concrete” (JGJ55-2011). The effects of SAP are different for different w/c ratios, especially for the healing approach. Lam and Hooton [31] reported that the concrete strength at different w/c ratios was approximately same. Lura [32] obtained that the loss rate of compressive strength made at the w/c ratio = 0.35 was 10‒20%. Esteves [33] found that the compressive strength was reduced by 15‒20% when the w/c ratio was 0.25, 0.3, and 0.35. Although a lower w/c being was used in previous studies, the values of w/c = 0.3 and w/c = 0.37 were selected to improve the volume stability and durability of the concrete. It was also obtained that the SAPs should be added at 0.0–1.0% of cement, as suggested by Mousavi [28]. SAP particles absorb water during mixing and reduce the concrete’s workability. Comparing mixtures with different workability is not accurate, based on the energy needed for compacting the specimens. Hence, additional water needed to be added to the concrete mixture, and this could be determined using the methods suggested by Mousavi and Powers [28,34]. The mix proportion scheme of the cement concrete is shown in Table 2. The combination mode of all the test numbers and test scheme of portland cement concrete are presented in Table 3.

The mechanical properties such as compressive strength, flexural strength, elastic modulus, and splitting tensile strength were analyzed, refer to “the standard test method of mechanical properties on ordinary concrete” (GB/T50081-2002). And the total of 3 cubes (150 mm × 150 mm × 150 mm in size) were tested under standard conditions, and the compressive strength test was performed at 1, 7, 14, and 28 days. The standard prism of 150 mm × 150 mm × 300 mm was measured by a Tm-2 elastic modulus tester. The dry shrinkage, autogenous shrinkage, and durability of the concrete at 3, 7, 14, and 28 days were investigated by the non-contact method. Three prismatic samples (100 mm × 100 mm × 515 mm in size) were measured in each batch. When preparing the test mold in the non-contact experiment, lubricating oil and two layers of plastic film were needed to line the test mold. It was necessary to apply lubricating oil evenly on the contact surface between the film and the test mold, and fix the reflection target at both ends of the test mold. After pouring the concrete mixture into the trial mold, it needs to be vibrated and smoothed, and a plastic film should be used for securing the sample against water loss through its surfaces. Then, the specimens were positioned under constant humidity conditions of (60 ± 5)% and a temperature of (20 ± 2) °C. The initial setting time of the concrete was measured when the specimen was formed. When the concrete was initially condensed, the initial readings on both sides of the specimen were measured and read. Then, the deformation readings on both sides of the specimen were measured at intervals of 6 h. The position and direction of the specimen placed on the deformation tester always remained fixed during the whole process. In addition, a DTL-6 concrete chloride ion electric flux tester was used to carry out an anti-chloride ion penetration, and each batch was conducted in 3 groups. The frost resistance of the concrete was determined by a KDR-V9 fast freeze‒thaw cycle. Scanning electron microscopy (SEM) tests were conducted to investigate the concrete microstructure.

The basic mechanical properties, volume stability, and frost resistance of concrete are included in the specific test process (see Figure 2).

## 3. Results and Discussions

The effects of SAP content, particle size, and water‒cement ratio on the mechanical performances, volume stability, and durability of concrete were studied through laboratory tests, thus providing theoretical and technical support for practical engineering applications [35].

### 3.1. Mechanical Properties

#### 3.1.1. Compressive Strength

The compressive strength of concrete aged for 1, 3, 7, 14, and 28 days was measured under standard conditions of curing. The compressive strength of concrete at two water‒cement ratios is shown in Figure 3, from which we found that the compressive strength was directly proportional to the age and water‒cement ratio, which was lower than that of the control group. On the one hand, the compressive strength of the concrete was reduced by SAP, due to the rapid development of an early hydration process of cement. On the other hand, SAP, a soft microcapsule, may reduce the compressive strength of concrete, because the SAP caused size expansion after absorbing water and formed more pores [23]. The upper and lower limits existed in the relationship between compressive strength and age. The compressive strength at a w/c ration of 0.3 was encompassed by the area of y = 1.538 + 12.5 and y = 1.18x + 27.95, and when the water‒cement ratio was 0.37, it was encompassed by the region of y = 1.846x + 10 and y = 1.45 + 33.64 at a w/c ration of 0.37. When the water‒cement ratio and SAP content were fixed, the compressive strength first increased and then decreased with an increase in SAP particle size. The long-term performance of concrete was significantly affected by different SAP particle sizes. However, the impact of the particle size on the loss of concrete strength is controversial. Mousavi [28] found that small particle size can lead to more pores in concrete, while Snoeck [36] reported that small particle size can lead to greater strength loss. Meanwhile, this paper concluded reductions of 9.04%, 5.00%, and 13.60% in the compressive strength for 0.15-, 0.3-, and 0.6-mm SAP particles, respectively. This could be explained by the fact that the SAP particle size affected the pore structure, and the porosity size influenced the strength [24,37]. When the SAP particle size was 0.3 mm, more pores of less than 800 microns existed in the concrete, which mainly affected the concrete strength. In addition, the SAP chemistry has a significant impact on the mechanical characteristics [28]. The effects of the SAP particles with different chemistry adopted in this paper on the curing approach of concrete were different, which could also be explained by this unusual trend. The best particle size of superabsorbent SAP was 0.3 mm, and its compressive strength was closest to the control group. The compressive strength at the two w/c ratios of 0.3 and 0.37 was reduced by 9.89% and 9.21%, respectively. The loss rate in the compressive strength was smaller than the result of 50% and 24% found in research by Lam and Ding [31,38]. It was, of course, also different that the concrete strength with 0.3% and 0.6% mixtures was reduced by 4% and 20%, respectively, as reported by Mechtcherine [39]. However, this paper was closest to the results of Piérard’s study, which observed concrete strength reduction by 7% and 13%, respectively [40]. This was because of the difference in the amount of additional water and SAP.

The SAP particle size, content, and 28-day concrete compressive strength were, respectively, set as the *X*-axis, *Y*-axis, and *Z*-axis, refer to Figure 4. To construct a prediction equation for the 28-day concrete compressive strength, a three-dimensional surface was used for fitting.

The Chebyshev 2D model was adopted, and the specific equation was Formula (1):Z = Z_0_ + A_1_cos(acos(x)) + A_2_cos(2acos(x)) + B_1_cos(acos(y)) + B_2_cos(2acos(y)) + C_1_cos(acos(x))cos(acos(y))(1)

The specifically fitted constants are presented in Table 4, which shows that the correlation coefficients R^2^ at the w/c ratios of 0.3 and 0.37 were 0.843 and 0.912, respectively. This indicated that the difference between the predicted and measured value was small, which further verified the rationality of the prediction equation. The standard error is abbreviated as SE in Table 4.

#### 3.1.2. Flexural Strength

Flexural strength is an important performance parameter of concrete, which reflects the concrete’s ability to resist failure under shear. Therefore, the influence of the SAP content and particle size on the flexural strength of concrete is discussed in this section.

The 28-day flexural strength of concrete at two water‒cement ratios is shown in Figure 5. The effect of the particle size on the flexural strength was greater than that of the SAP content. From Figure 5a, the overall flexural strength of the concrete with SAP added, at the same mix ratio, was lower than that of the control group. It was also inversely proportional to the SAP content, which finding was consistent with the research by Assmann [11]. The SAP swelled greatly when in contact with water in the early stage of the concrete hydration reaction. As the concrete hydration reaction deepened, most of the water in the SAP was sucked away, leaving cavities inside the concrete. Then, the concrete volume occupied by the cavities also increased with the increase in the SAP content. This caused the splitting tensile strength to decrease with the increase in the SAP content [38]. The smaller the w/c ratio, the greater the loss of flexural strength. From Figure 5b, the flexural strength first increased and then decreased with the increase in the SAP particle size, which is because the SAP chemistry has a significant impact on the mechanical characteristics. The reason for this unusual trend was that the effects of the SAP particles with different chemistry adopted in this paper on the curing of the concrete were different. The optimal particle size of SAP was 0.3 mm when the w/c ratio and mixing amount were constant, and its flexural strength value was closer to the control group.

The Chebyshev2D, a two-dimensional nonlinear surface model, was used to fit the relationship among the particle size, the content of SAP, and flexural strength, which is shown in Figure 6. The fitting relationship between the three under the two w/c ratios is shown in Formula (1), and the constants for the specific fitting equation are shown in Table 5. From Figure 6 and Table 5, there was not much difference between the predicted and measured flexural strength. The correlation coefficient R^2^ was greater than 0.95, which indicated the rationality of the prediction equation.

#### 3.1.3. Elastic Modulus

The elastic modulus is a necessary parameter for calculating the crack development, structure deformation, and temperature stress of concrete [39,40]. The calculation of the elastic modulus can be obtained by Formula (2):(2)$$E_{c}=\frac{p_{2}-p_{1}}{A}\times \frac{L}{\Delta L}$$
where *E_c_* is the elastic modulus, *p*_1_ is the load at 0.5 MPa stress (N), *p*_2_ is the ultimate failure load of 40% (N), *A* is the area (mm^2^), *L* is the gauge length for measuring the deformation, (mm), and ∆*L* is the deformation value of the specimen under *p*_1_ to *p*_2_ load(mm).

The calculated elastic modulus of concrete aged for 28 days under different water‒cement ratios is plotted in Figure 7. The SAP reduced the elastic modulus of the concrete, which was less than that of the control group under the two water‒cement ratios. The SAP content and elastic modulus of the concrete are inversely proportional. The elastic modulus first increased and then decreased with an increase in the SAP particle size. Compared with the control group, the elastic modulus decreased by 15.58% and 12.98%, respectively, when the w/c ratio was 0.3 and 0.37. This was because the influence of the SAP on the elastic modulus of the concrete was mainly determined by the pore structure of the hardened concrete and the degree of hydration of cementitious materials. The SAP could increase the porosity of the concrete, so the elastic modulus was reduced compared to the control group [41,42].

The relationship among the SAP particle size, content, and elastic modulus under different water–cement ratios is presented in Figure 8. The fitting prediction equation for the 28-day concrete elastic modulus is presented in Table 6. From Figure 8 and Table 6, the correlation coefficients R^2^ of the predicted and measured values are 0.956 and 0.974, respectively, under two w/c ratios, which further verifies the rationality of the fitting equation.

#### 3.1.4. Splitting Tensile Strength

It is, of course, important to know the splitting tensile strength to determine the cracking resistance of concrete in structural design. The calculation of the splitting tensile strength of the concrete cube is shown in Formula (3):(3)$$f=\frac{2p^{\prime }}{\pi A^{\prime }}$$
where *f* is the splitting tensile strength, (MPa), *p’* is the load, (N), and *A’* is the area (mm^2^).

The splitting tensile strength of concrete, under two water–cement ratios, is given in Figure 9, which shows that the SAP content and splitting tensile strength of the concrete were inversely proportional when the water‒cement ratio and particle size were constant. This was because the SAP was very water-absorbent and swelled with water at the beginning of the hydration reaction of the concrete. With the strengthening of the hydration reaction, the water inside the SAP was sucked away by the surrounding medium to leave pores. The greater the SAP content, the larger the concrete pores. The splitting tensile strength of concrete tended to increase firstly and then decrease with an increase in the particle size. The loss of concrete splitting tensile strength at the w/c ratio of 0.3 and 0.7 was 10.23% and 17.64%, respectively, which was similar to the research results of Mechtcherine [39].

The fitting diagram of the SAP particle size, content, and splitting tensile strength of concrete aged for 28 days is shown in Figure 10, and the fitting constant of the prediction equation of concrete splitting tensile strength is presented in Table 7. From Figure 10 and Table 7, the correlation coefficients at the two water‒cement ratios were 0.990 and 0.986, respectively, which were closer to 1. Thus, this indicated that the predicted splitting tensile strength was closer to the measured value.

### 3.2. Volume Stability

Early cracking of concrete often occurs in windy, dry, and high-temperature areas. The concrete cracking can be effectively solved by the internal curing material called SAP, which has broad application prospects [43]. Its service life and strength can be affected by the volume stability of the concrete, primarily regarding its shrinkage performance [22,44]. The impacts of the SAP on the shrinkage and drying shrinkage of concrete are mainly discussed in this section. To simplify the analysis, the impact of the SAP particle size on the volume stability was studied with a fixed w/c ratio of 0.37.

#### 3.2.1. Drying Shrinkage

The drying shrinkage of SAP concrete at different ages is demonstrated in Figure 11. From Figure 11, it can be seen the SAP content was inversely proportional to the drying shrinkage of the concrete when the age was the same. The drying shrinkage of concrete was improved by 18.79% and 15.23%, respectively, when the age of the concrete was 28 days and 56 days. When the mixing amount was constant, the drying shrinkage of the concrete first decreased and then increased with the SAP particle size, and 0.3 mm was selected as the optimal SAP particle size. There can be a significant improvement of the drying shrinkage performance of concrete with an SAP content of 0.4% when taking into account the adverse effect of SAP on the concrete’s mechanical properties. Recent results from research by Assmann [11] and Justs [5] suggest that the autogenous shrinkage, drying shrinkage, and tensile creep of concrete can be effectively improved by SAP. The result that the self-shrinkage and crack resistance of HPC and UHPC could be reduced by SAP was also found by Shen [45]. Recent results from their research were consistent with this article, but there were certain differences in the improvement effect. However, the reduction of autogenous shrinkage was about 10% greater than the results of this paper, which could be explained by the different water‒cement ratios and water addition [45].

#### 3.2.2. Autogenous Shrinkage

The autogenous shrinkage of concrete at different ages with a fixed w/c ratio of 0.37 is shown in Figure 12, which indicates that the SAP content and the autogenous shrinkage of the concrete were inversely proportional. The impact of the SAP on the concrete autogenous shrinkage became greater when the age was greater than 7 days. The autogenous shrinkage of the concrete with an SAP content of 0.2%, 0.4%, and 0.6% was reduced by 5.68%, 44.28%, and 62.38%, respectively, when the SAP particle size was 0.3 mm. However, the autogenous shrinkage of concrete can be greatly improved with 0.4% SAP content, when taking into account the adverse effect of SAP on the mechanical properties of the concrete. In addition, this shrinkage showed a trend of first decreasing and then increasing with an increase in the SAP particle size. When the age was greater than 7 days, the impact of the SAP particle size on the autogenous shrinkage of the concrete became greater. It has been found in related research that some of the autogenous shrinkage of concrete could be offset by the SAP during the accelerated period of cement hydration. Moreover, the reduction rate of the concrete’s autogenous shrinkage in this paper was basically between 32% and 44% of the values in the study by Mechtcherine [7,39]. This was because SAP in cement pore solutions played a major role in the mitigation of autogenous shrinkage. In addition, the shrinkage of concrete could be significantly reduced by an increase in optical fiber, as studied by Fan [46].

### 3.3. Durability

#### 3.3.1. Frost Resistance

The serviceability of concrete in a freeze-thaw environment is determined by its frost resistance. With SAP, a certain closed pore structure inside the concrete after the water release is formed, so the frost resistance of the concrete can be improved [20,47].

The effect of SAP on the relative dynamic elastic modulus of concrete when the water–cement ratio is 0.37 is shown in Figure 13. Compared with the control group, the relative dynamic elastic modulus of concrete after 25 to 100 cycles was reduced by 15.43%, 15.13%, 13.93%, and 12.39%, respectively, taking the SAP content of 0.4% as an example. So, the frost resistance of concrete is significantly improved with the addition of SAP. This was because the SAP promoted the later hydration reaction and improved the compactness of the concrete. On the other hand, the greater internal porosity of concrete improved the frost resistance of the concrete [26].

The impact of SAP on the mass loss rate of concrete when the water‒cement ratio is 0.37 is shown in Figure 14, which makes clear the positive correlation between the mass loss rate of the concrete and the number of cycles. From Figure 14b, the mass loss rate of the concrete with the SAP particle size of 0.3 mm was relatively low when the amount of SAP was 0.4%. It was established that the SAP particle size of 0.3 mm could effectively improve the frost resistance of concrete. Laustsen [26] also found that the voids created by SAP could protect concrete from freezing, just like other voids. One of the conclusions of Beushause et al. [20] was that enough voids created by SAP could provide sufficient frost resistance for concrete. In other words, the durability and frost resistance of HPC, UHPC, and cement mortar can be effectively improved by SAP.

#### 3.3.2. Chloride Penetration Resistance

Concrete foundations or enclosure structures that are exposed to chloride ion corrosion environments for a long time will affect the service life of the concrete [2,21,24].

The effect of SAP on the chloride resistance of concrete is presented in Figure 15, which shows that the concrete electric flux and the amount of SAP are inversely proportional. The larger the SAP content, the smaller the electric flux, which meant that the chloride ion penetration resistance of the concrete increased. The electric flux of concrete with 0.2%, 0.4% and 0.6% SAP content decreased by 10%, 20% and 21% than the control group, respectively. In addition, there was a trend of first increasing and then decreasing with an increase in the SAP particle size for the chloride ion penetration resistance of concrete when the SAP content was constant. When the SAP particle size was 0.3mm, the concrete electric flux decreased the most compared to the control sample, which was 31%. Therefore, when the SAP particle size was 0.3 mm, it could effectively improve the corrosion resistance of the concrete. Recent results from research by Dang [3] show that the carbonation resistance and chloride penetration resistance of concrete can be improved by pre-absorbed SAP. This gives a further indication of how SAP can improve the durability of concrete.

### 3.4. Characteristics of Microscopic Pore Structure of Concrete

Concrete is a typical porous medium material with a wide pore size distribution, and a complex and diverse pore structure [26]. The pore size ranges from micro to macro scale, and the pores of the material have an important influence on its macro performance [48,49]. Therefore, the concrete microstructure and the impact of SAP on the mechanical performance of concrete were studied by SEM in this section.

#### 3.4.1. Calculation of Pore Structure Parameters

Three main indicators were included in the stomatal structure parameters, which were the mean radius of the pore, the stomatal volume percentage, and the coefficient of average pore spacing [23].

Porosity

The porosity was calculated as Formula (4):(4)$$\beta =\frac{S_{p}}{S_{0}}$$
where *β* is the porosity, *S_p_* is the porosity area, and *S*_0_ is the cross-sectional area of the sample.

2.Equivalent pore diameter

According to Delesse’s law, the area fraction of pores can be regarded as an unbiased expected value of bulk density. Since the pores are not perfectly round, the equivalent diameter of the pores is used to express the size of the pores, as shown in Formula (5):(5)$$D_{e}=2\sqrt{\frac{S}{\pi }}$$
where *D*_e_ is the equivalent pore diameter, and *S* is the coefficient of average pore spacing.

3.Coefficient of average pore spacing

Using Emmanuel’s formula, the coefficient of average stomatal spacing was calculated as Formula (6):(6)$$S=\frac{2FP^{2}}{aA}$$
where *S* is the coefficient of average pore spacing, *F* is the volume percentage of effective slurry, *P* is the volume percentage of cement paste, *a* is the specific surface area of pores, and *A* is the pore volume percentage.

#### 3.4.2. Porosity

The SEM image and the influence of SAP on the concrete porosity under different water‒cement ratios are shown in Figure 16. When SAP was added to the concrete, the concrete gel pores and small pores increased, and the sizes of large and medium pores decreased, which refined the concrete pore structure (see Figure 16a). At the same time, the later release of water from the SAP promoted secondary hydration of the cement and improved the compactness of the concrete. Eventually, the volume stability and durability of the concrete were improved [3]. From Figure 16b, it was shown that the porosity of the concrete was positively and negatively correlated with the SAP content and particle size, respectively. There was also a positive correlation between the w/c ratio and the concrete porosity. The concrete porosity was increased, and the thermal insulation performance of the concrete was improved, but concrete strength was reduced. Due to the formation of pores, the fluidity was reduced and the air content was increased by the SAP, thereby reducing the strength of the concrete. Moreover, Yu [41] showed that larger voids were formed by desorbed SAP, which harmed the cube strength of the concrete. The porosity of concrete with an SAP content of 0.2%, 0.4, and 0.6% increased by 22.25%, 30.08%, and 37.69% than the control group, respectively, but the strength of the concrete was reduced by 8.16%, 10.28%, and 14.02%, respectively. In addition, the porosity decreased with an increase in particle size. However, the porosity of some of the mixed-proportion concrete was less than that of the original control group, because the SAP could swell when in contact with water, increasing the porosity of the concrete. Thus, it is necessary to study the porosity classification further.

To further study the relationship between porosity and concrete strength, the porosity was divided into 5 grades: 10–200 μm, 200–600 μm, 600–1000 μm, 1000–1400 μm, and 1400–1600 μm. The different porosity classifications are shown in Figure 17a. From Figure 17a, one of the conclusions was that the main part of the concrete had a porosity of 10‒1000 μm, accounting for 79.11% of the total porosity on average, while the porosity gradings of 10–200 μm, 200–600 μm, and 600–1000 μm accounted for 36.59%, 25.85%, and 16.67% of the total porosity, respectively. From Figure 17b, the porosity classification was divided into three grades to analyze its impact on the concrete strength. It was found that the correlation coefficients R^2^ of the porosity grading at 10–200 μm, 200–1000 μm, and 1000–1600 μm were 0.413, 0.383, and 0.321, respectively. Thus, it was concluded that a porosity classification of fewer than 1000 μm had a great influence on the concrete strength. This was because the gel pores and small pores increased when SAP was added to the concrete. The size of the large and medium pores decreased, which was consistent with the results of Dang [3].

#### 3.4.3. Mean Radius of the Pore

The effects of the SAP content and particle size on the mean radius of concrete pores are shown in Figure 18. From Figure 18a, it can be seen that a high water‒cement ratio had a significant effect on the mean radius of pores. The average pore radius of the concrete at the w/c ratio of 0.37 with an SAP content of 0.2%, 0.4%, and 0.6% was 39.55%, 56.82%, and 73.61%, respectively, which was higher than that of the control group. In addition, it was easier for the higher water‒cement ratio and larger particle size to introduce an increase in the average radius of the pores. From Figure 18b, it can be seen that the mean radius of the concrete pores was negatively correlated with its strength. This was because the larger the mean radius of concrete pores, the wider the interface transition zone would be. Thus, the microhardness of the concrete was lower and then resulted in the compressive strength of concrete becoming small [25].

#### 3.4.4. Coefficient of Average Pore Spacing

Figure 19 presents the average spacing coefficient of concrete pores with different mix ratios and its relationship with compressive strength. From Figure 19a, it can be seen that there was a positive correlation between the SAP content and particle size and the average spacing coefficient of the concrete pores. The average spacing coefficient of the pores of the concrete with an SAP content of 0.2%, 0.4%, and 0.6% was increased by 61.57%, 134.86%, and 198%, respectively. Moreover, when the SAP particle size increased from 0.15 mm to 0.3 mm, the average pore spacing coefficient increased by 12.22%. In addition, particle size possessed a greater impact on the average spacing coefficient of concrete pores when the w/c ratio was large. It had little effect on the coefficient of average pore spacing for an SAP particle size greater than 0.3 mm. In Figure 19b, a negative correlation is shown for the relationship between the average spacing coefficient of the concrete pores and its strength. When the total porosity of the concrete was similar, the more reasonable the pore size distribution was, and the higher the concrete strength. That is, the smaller the average spacing coefficient, the higher the concrete strength [48,49].

## 4. Conclusions

This paper studies the influence of SAP on the mechanical performances, volume stability, and durability of concrete, based on laboratory experiments, and the characteristics of the microscopic pore structure of concrete have been further explored. Some conclusions are as follows.

There are inversely proportional relationships among the SAP content and the concrete mechanical performances, such as compressive strength, flexural strength, elastic modulus, and splitting tensile strength. The mechanical properties of the concrete tend to first increase and then decrease with the SAP particle size. In addition, the mechanical performances of concrete mixed with SAP can be reduced by 10% on average. The best SAP particle size and content are 0.3 mm and 0.2%, respectively, which have the most comprehensive effect to ensure the mechanical properties of concrete.Through the Chebyshev2D model adopted, the relationship among the SAP particle size, content and mechanical performances at different water‒cement ratios have been obtained. This provides great assistance with the prediction of concrete’s mechanical properties.There is an inversely proportional relationship between the SAP content and shrinkage of concrete. The volume stability of the concrete tends to first increase and then decrease with SAP particle size. The drying shrinkage and autogenous shrinkage of concrete can be best improved by 16.09% and 30.62%, respectively, and thus, the volume stability of concrete is improved.A certain closed pore structure is formed inside the concrete after the SAP releases water. Therefore, SAP can effectively improve the frost resistance of concrete by 28.3%, and the resistance to chloride ion penetration by 16.6%. Overall, the frost resistance and corrosion resistance of the concrete are improved.Through the SEM research on the microscopic pore structure characteristics of SAP concrete, it has been found that the mix proportion has a significant effect on the pore structure parameters. It is also seen that the development of concrete compressive strength is mainly affected by a pore classification of fewer than 1000 μm, and the order of importance of the graded porosity has been obtained. It is determined by the use principle for SAP, which involves a low water‒cement ratio, coarse particle size, and low content. However, the optimal use principle for SAP still needs to be determined through more experimental studies, which are necessary for future work to confirm the database.

## Figures and Tables

**Figure 1 materials-14-03232-f001:**
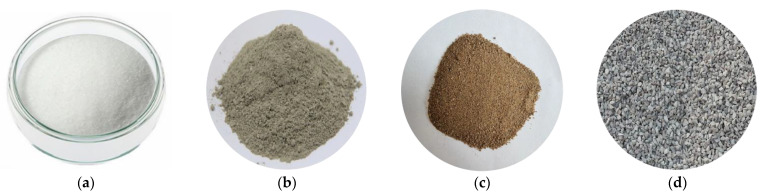
Test materials: (**a**) SAP; (**b**) cement; (**c**) fine aggregate; (**d**) coarse aggregate.

**Figure 2 materials-14-03232-f002:**
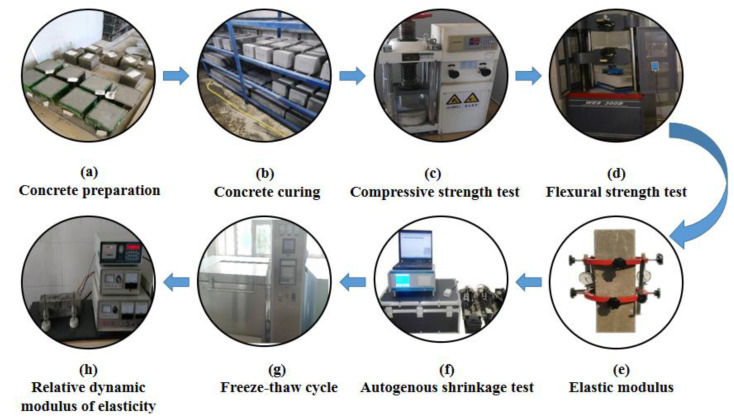
The process of the main test.

**Figure 3 materials-14-03232-f003:**
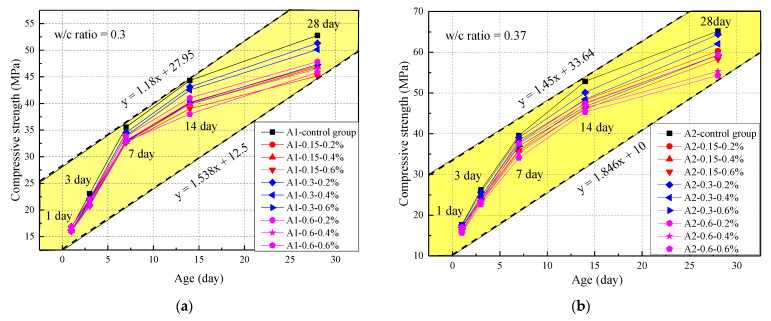
Concrete compressive strength under different water–cement ratios: (**a**) water–cement ratio of 0.3; (**b**) water–cement ratio of 0.37.

**Figure 4 materials-14-03232-f004:**
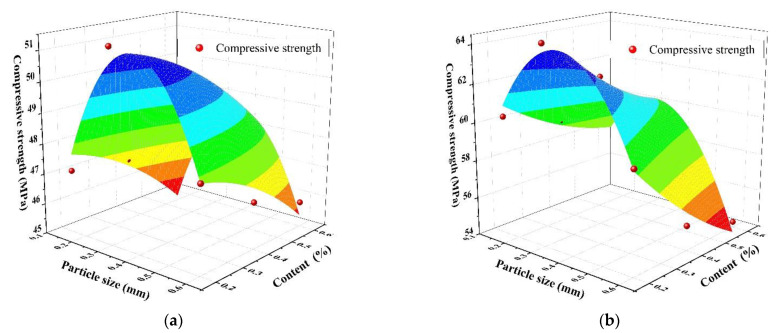
The relationship between SAP particle size, content and compressive strength: (**a**) water–cement ratio of 0.3; (**b**) water–cement ratio of 0.37.

**Figure 5 materials-14-03232-f005:**
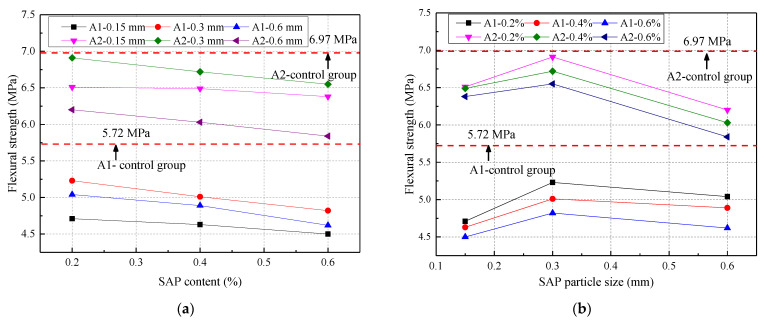
Flexural strength under different water–cement ratios: (**a**) different SAP content; (**b**) different SAP particle size.

**Figure 6 materials-14-03232-f006:**
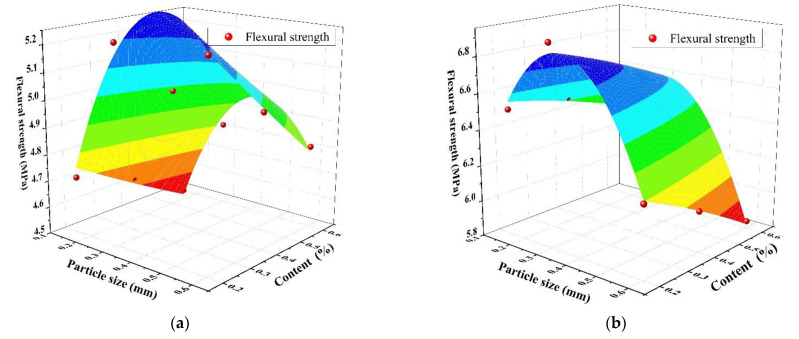
The relationship among SAP particle size, content, and flexural strength: (**a**) water–cement ratio of 0.3; (**b**) water–cement ratio of 0.37.

**Figure 7 materials-14-03232-f007:**
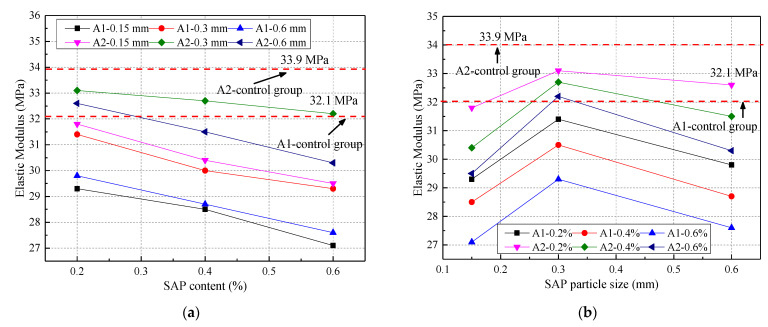
Elastic modulus of concrete under two water–cement ratios: (**a**) different SAP content; (**b**) different SAP particle size.

**Figure 8 materials-14-03232-f008:**
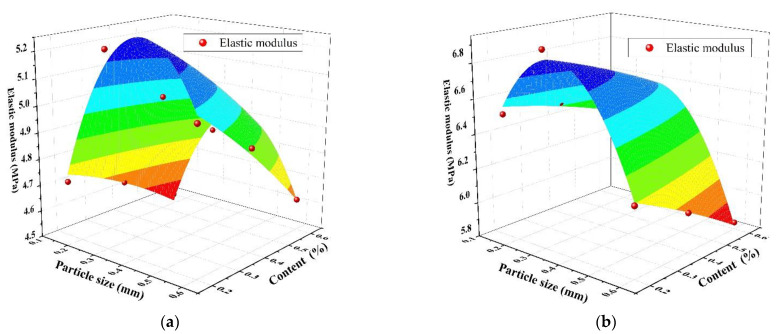
The relationship between SAP particle size, content, and elastic modulus: (**a**) water–cement ratio of 0.3; (**b**) water–cement ratio of 0.37.

**Figure 9 materials-14-03232-f009:**
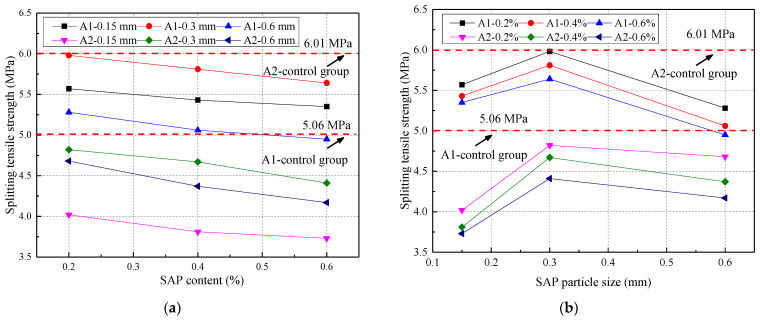
Splitting tensile strength of concrete under two water–cement ratios: (**a**) different SAP content; (**b**) different SAP particle size.

**Figure 10 materials-14-03232-f010:**
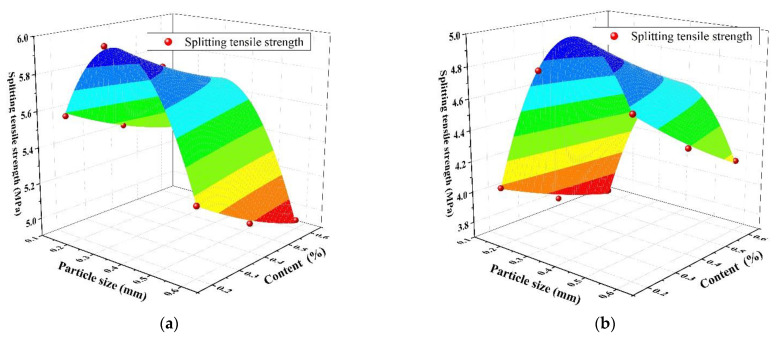
The relationship between SAP particle size, content, and splitting tensile strength: (**a**) water–cement ratio of 0.3; (**b**) water–cement ratio of 0.37.

**Figure 11 materials-14-03232-f011:**
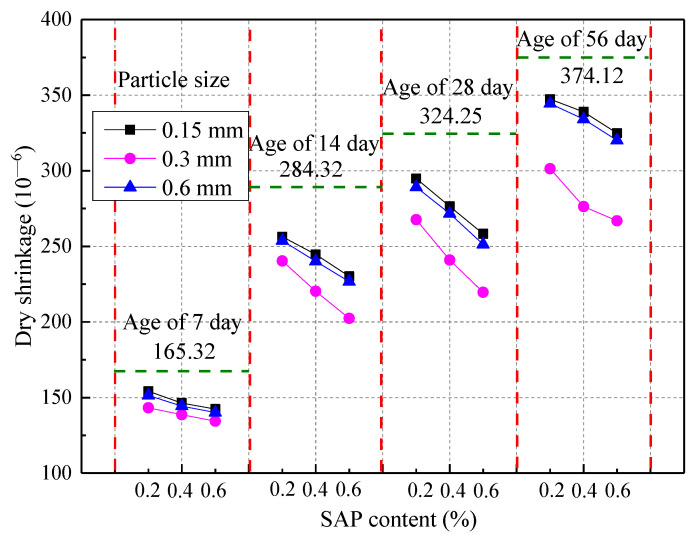
Dry shrinkage of SAP concrete at different ages.

**Figure 12 materials-14-03232-f012:**
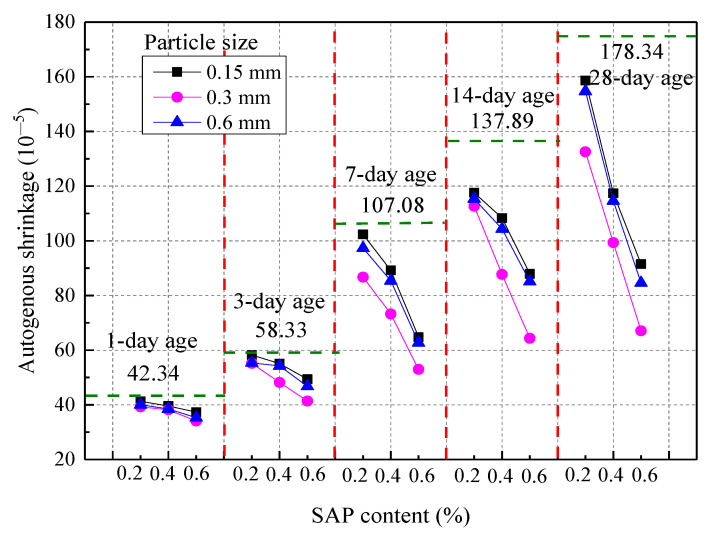
Autogenous shrinkage of concrete at different ages.

**Figure 13 materials-14-03232-f013:**
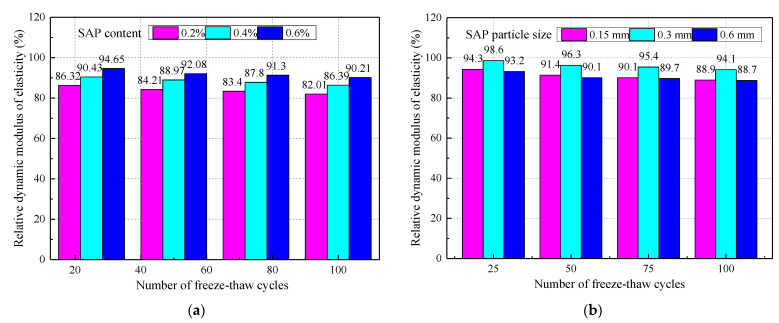
The influence of SAP on the relative dynamic elastic modulus of concrete: (**a**) SAP particle size of 0.3 mm; (**b**) SAP content of 0.4%.

**Figure 14 materials-14-03232-f014:**
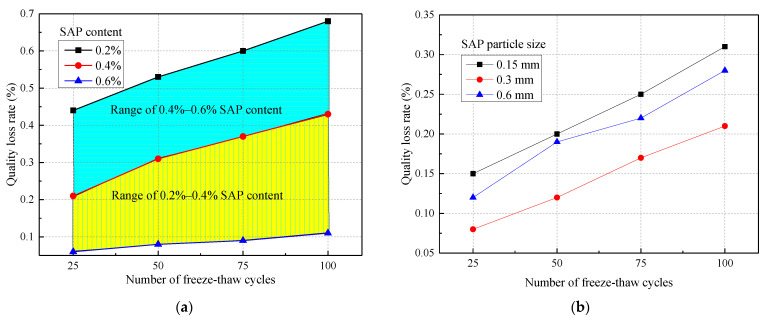
Effect of SAP on concrete mass loss rate: (**a**) SAP particle size of 0.3 mm; (**b**) SAP content of 0.4%.

**Figure 15 materials-14-03232-f015:**
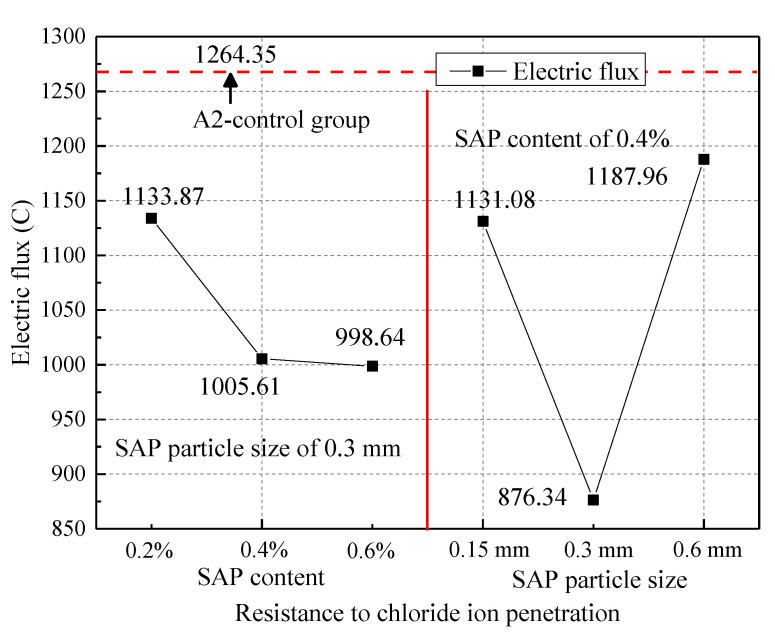
The influence of SAP on the ability of concrete to resist chloride ions.

**Figure 16 materials-14-03232-f016:**
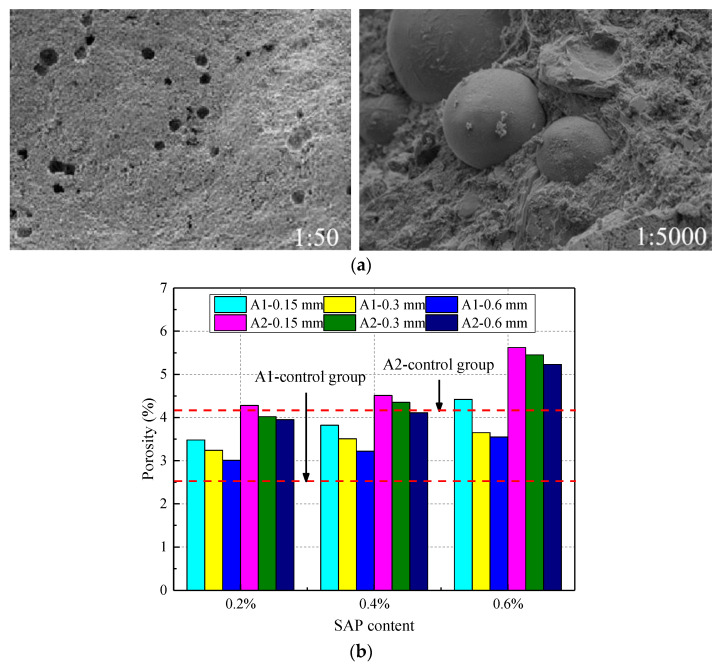
The porosity of concrete under different water–cement ratios: (**a**) SEM image of concrete with addition of SAP; Reprinted from ref. [3]. (**b**) different SAP contents.

**Figure 17 materials-14-03232-f017:**
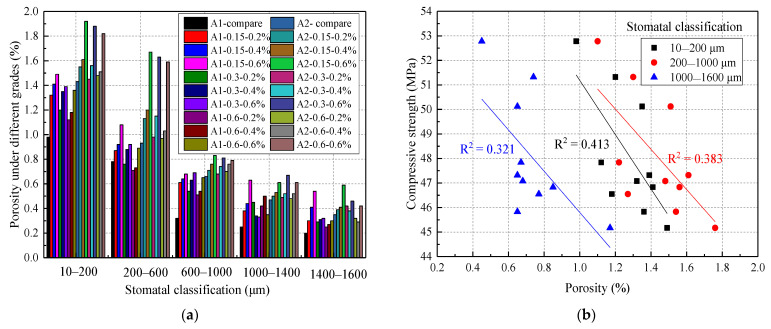
The effect of pore grading on concrete strength: (**a**) different graded porosity; (**b**) the relationship between porosity and compressive strength.

**Figure 18 materials-14-03232-f018:**
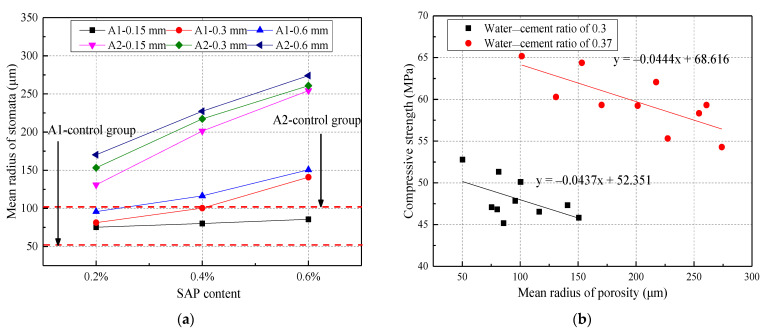
The effect of SAP on the mean radius of stomata under different water–cement ratios: (**a**) different SAP content; (**b**) the relationship between average pore radius and compressive strength.

**Figure 19 materials-14-03232-f019:**
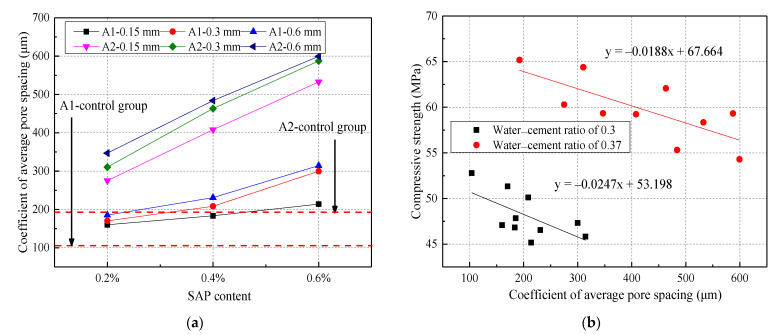
The effect of SAP on the coefficient of average pore spacing under different water–cement ratios: (**a**) different SAP content; (**b**) relationship between the coefficient of average pore spacing and compressive strength.

**Table 1 materials-14-03232-t001:** Basic performance indicators of SAP and cement. Adapted from ref. [3].

Materials	Index	30 mesh	50 mesh		100 mesh
SAP	Particle size (mm)	0.6	0.3		0.15
	Water absorption rate	200	400		700
	Specific gravity	1.52	1.51		1.48
	d50	315	162		46
	Water absorption in deionized water (g/g)	212	335		725
	Water absorption in cement hydration solution (g/g)	32	53		74
	Bulk density (g/mL)	0.6–0.8			
Cement	Density (kg/m^3^)	3150			
	Initial setting time (min)	200			
	Final setting time (min)	275			
	Compressive strength (MPa)	3-day		28.5	
		28-day		51.4	
	Flexural strength (MPa)	3-day		5.8	
		28-day		8.8	

**Table 2 materials-14-03232-t002:** Mix proportion of cement concrete.

Item	Water–Cement Ratio	Material Consumption of Cubic Concrete (kg/m^3^)				
		Cement	Sand	Pebble	Water	Additional Water
A1	0.3	550	705	1040	165	29.7
A2	0.37	419	820	1030	155	20.95

**Table 3 materials-14-03232-t003:** Test scheme of Portland cement concrete.

Item	w/c Ratio	SAP Particle Size (mm)	SAP Content (%)	Item	w/c Ratio	SAP Particle Size (mm)	SAP Content (%)
A1-control	0.3	0	0	A2-control	0.37	0	0
A1-0.15-0.2%	0.3	0.15	0.2	A2-0.15-0.2%	0.37	0.15	0.2
A1-0.15-0.4%		0.15	0.4	A2-0.15-0.4%		0.15	0.4
A1-0.15-0.6%		0.15	0.6	A2-0.15-0.6%		0.15	0.6
A1-0.3-0.2%	0.3	0.3	0.2	A2-0.3-0.2%	0.37	0.3	0.2
A1-0.3-0.4%		0.3	0.4	A2-0.3-0.4%		0.3	0.4
A1-0.3-0.6%		0.3	0.6	A2-0.3-0.6%		0.3	0.6
A1-0.6-0.2%	0.3	0.6	0.2	A2-0.6-0.2%	0.37	0.6	0.2
A1-0.6-0.4%		0.6	0.4	A2-0.6-0.4%		0.6	0.4
A1-0.6-0.6%		0.6	0.6	A2-0.6-0.6%		0.6	0.6

Note: the item is expressed in the form of AX-Y-Z, where AX represents the mix proportion, Y is the SAP particle size, and Z is the amount of SAP added. The number 1 and 2 represent the water–cement ratio of 0.3 0.37, respectively.

**Table 4 materials-14-03232-t004:** Prediction equation fitting of 28-day concrete in compressive strength.

Water–Cement Ratio	Z_0_		A_1_		A_2_		B_1_		B_2_		C_1_		R^2^
	Value	SE	Value	SE	Value	SE	Value	SE	Value	SE	Value	SE	
0.3	2.055	11.748	51.754	10.352	−34.444	6.268	0.871	11.586	−5.083	6.924	1.893	8.547	0.843
0.37	21.806	13.703	59.687	12.074	−40.284	7.310	−13.867	13.514	5.604	8.076	−14.690	9.970	0.912

**Table 5 materials-14-03232-t005:** Prediction equation fitting of 28-day concrete in flexural strength.

Water–Cement Ratio	Z_0_		A_1_		A_2_		B_1_		B_2_		C_1_		R^2^
	Value	SE	Value	SE	Value	SE	Value	SE	Value	SE	Value	SE	
0.3	1.189	0.676	5.862	0.595	−3.025	0.361	−0.600	0.666	0.063	0.398	−1.071	0.492	0.969
0.37	1.093	0.803	6.338	0.707	−4.580	0.428	−0.025	0.791	−0.188	0.473	−1.095	0.584	0.974

**Table 6 materials-14-03232-t006:** Prediction equation fitting of concrete in elastic modulus.

Water–Cement Ratio	Z_0_		A_1_		A_2_		B_1_		B_2_		C_1_		R^2^
	Value	SE	Value	SE	Value	SE	Value	SE	Value	SE	Value	SE	
0.3	0.072	0.740	6.393	0.652	−3.642	0.395	−0.0458	0.730	−0.292	0.436	−1.012	0.539	0.956
0.37	1.093	0.803	6.338	0.707	−4.580	0.428	−0.025	0.791	−0.188	0.473	−1.095	0.584	0.974

**Table 7 materials-14-03232-t007:** Prediction equation fitting of concrete in splitting tensile strength.

Water–Cement Ratio	Z_0_		A_1_		A_2_		B_1_		B_2_		C_1_		R^2^
	Value	SE	Value	SE	Value	SE	Value	SE	Value	SE	Value	SE	
0.3	−0.023	0.493	7.383	0.434	−5.309	0.263	−1.129	0.486	0.354	0.291	0.512	0.359	0.990
0.37	−3.556	0.704	11.622	0.620	−6.617	0.376	−1.033	0.694	0.271	0.415	−1.167	0.512	0.986

## Data Availability

Data available on request.

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
