# Peer review of "Effect of Superabsorbent Polymer on the Mechanical Performance and Microstructure of Concrete"

_materials, 2021, doi:10.3390/ma14123232_

Round 1
Reviewer 1 Report
- The current study investigates the influence of adding a polymer with super absorbing capabilities on the mechanical performance of concrete. For this, the authors study the microstructure of the new hybrid concrete and evaluate the influence of polymer particle size and volume on the mechanical durability. The author found that shrinkage was reduced while the volume stability of concrete improved. In addition, the authors found that crack was reduced while issues such as chloride penetration and the freeze-thaw resistance were also improved.
- Please consider reviewing the abstract and highlight the novelty, major findings and conclusions.
- The authors keep mentioning “the mechanical properties” but this sentence is very generic and does not clearly tell the readers what are those “mechanical properties” which were analysed?
- After line 86 please answer the following question: What is the research gap did you find from the previous researchers in your field? Mention it properly. It will improve the strength of the article.
- Please consider combining Tables 1 and 2 in one table
- Please reference Table 2 if this data was not measured by the authors
- Remove section 2.2.1 title is does not have any useful meaning
- Why did the authors choose those specific mixing portions? Were they randomly chosen or according to industry standards/recommendations or something else?
- Please remove all these sub-sub section titles and just merge all in one larger section. For example just keep 2.2 and call it Test procedure or protocol..etc
- Also you use the word test so many times in those sections its confusing the readers of what exactly this section refers to
- Line 160 abuse of referencing, it is not clear why those references are added there? Please remove them or give full credit for each of those references. There is only one line behind them which does not make sense to add four reference for such a generic sentence only!
- Authors did not mention in the abstract that Compressive strength/ Flexural strength/Elastic modulus/Splitting tensile strength/ were measured or evaluated!
- Line 1701-171 needs a reference to support this finding and explain why as well
- Line 179-180 needs a reference to support this finding and explain why as well
- Line 213 “the same mix ratio, was lower than that of the control group” why? Please discuss and explain then support with references, just saying this is in agreement with x study is not sufficient
- Line 257-248 needs a reference to support this finding and explain why as well
- The authors just keep stating important findings from their experimental work but never make any effort to explain why it happened or compare it with past work in the open literature
- Line 312 “certain differences in the improvement effect existed” what are these differences please mention some or all of them to make it more clear for the reader
- The authors must provide a list of nomenclature for all the symbols and Greek letters mentioned in this manuscript at the end or the start of the manuscript
- Some of the results are merely described and is limited to comparing the experimental observation. The authors are encouraged to include more detailed discussion in each section and critically discuss the observations from this investigation with existing literature.
Author Response
Dear Editor and Reviewers
Thank you very much for your letter. The reviewers’ comments concerning our manuscript entitled “Effect of Superabsorbent Polymer on the Mechanical Performance and Microstructure of Concrete”. These comments are all valuable and very helpful for revising and improving our paper, as well as important guiding significance to our researches. We have studied comments carefully and have made corrections which we hope meet with approval. All changes are marked in the track changes version.
Best wished
Sincerely

Reviewer 2 Report
The present manuscript develops the effect of superabsorbent polymer on and microstructural properties of concrete. This seems interesting for the construction industry. However, the manuscript should be improved and relevant changes are required, taking into account the following comments:
- My major criticism is the manuscript is too long with many figures that provide information that is sometimes repetitive. However, microstructural properties are discussed and SEM images are not shown.
- Please, justify the originality/novel of the present manuscript in comparison with previous works.
- The title is not clear, too “and” is confused.
-Why these mixed proportions of cement concrete are used (Table 3)?
- How is the use of acrylic-based materials justified due to the environmental impact associated with these non-biodegradable polymers?
- Please include standard deviation in the results in order to assess the significant differences among different samples.
- According to the evaluation of the properties (mechanical, microstructural, stability, etc.), which ones do the authors consider the optimal ones for the development of concrete that can substitute the current ones?
Author Response

(The authors gave the same response as above.)

Reviewer 3 Report
Manuscript ID: materials-1238495 Title: Effect of Superabsorbent Polymer on the Physical and Mechanical Properties and Microstructure of Concrete
Comments:
The article presents a very interesting experiment.
Materials and methods:
The materials and methods are fairly well described. Places for improvement have been marked in the text.
In the case of shrinkage testing, please describe in detail the sample preparation and the seasoning conditions during the measurements.
This is very important when determining autogenous shrinkage and drying shrinkage.

Author Response

(The authors gave the same response as above.)

Reviewer 4 Report
This paper experimentally studied the influence of SAP (percent and particle size) on the mechanical and durability properties of concrete. Results are interesting, while important details of SAP particles and SAP concrete mixtures were not considered in the manuscript. These details are so important. A major revision is necessary as follows:
Page 1, Lines 42-43: please add references.
Page 2, Lines 54-55: Please remove the phrase “home and abroad”.
Page 2, Lines 87-98: please highlight the limitations and research gaps of the literature and specifically determine the objectives of this work.
Page 3, Lines 101-104 and Table 1: Did you measure the water absorption of SAP in pore solution? This parameter is so important for the concrete mixture. Please measure both water absorption in deionized water and pore solution. Different methods are explained in the following reference:
[-] Mousavi, S. S., Ouellet-Plamondon, C. M., Guizani, L., Bhojaraju, C., & Brial, V. (2020). On mitigating rebar–concrete interface damages due to the pre-cracking phenomena using superabsorbent polymers. Construction and Building Materials, 253, 119181.
Table 1: please mention more details of SAP used in the experimental program, including specific gravity, bulk density, d50, SAP chemistry, production technique, and particle shape.
Page 3, Lines 124-125: why w/c ratios of 0.30 and 0.37 were selected for this study? SAP effects are different for different w/c ratios, especially for the healing approach. Please compare your results with the literature. It seems that the authors ignored many researches in this field. Results should be compared with the previous studies.
Table 3: Please add more details regarding SAP concrete. SAP particles absorb water during mixing and reduce the concrete workability. Comparing mixtures with different workability is not accurate based on the energy needed for compacting the specimens. Hence, more water (additional water) is needed to add to the concrete mixture. Previous studies severally confirmed this approach. Additional water was considered in the present study? If yes please mention in Table 3, If no, please mention the slump flow of the mixtures.
Table 4: Please change “mix ratio” to “w/c ratio”.
Fig. 3: Results regarding particle size are interesting but needed more explanations in this study. Please compare your results with the experimental result reported by the following reference:
[-] Mousavi, S. S., Ouellet-Plamondon, C. M., Guizani, L., Bhojaraju, C., & Brial, V. (2020). On mitigating rebar–concrete interface damages due to the pre-cracking phenomena using superabsorbent polymers. Construction and Building Materials, 253, 119181.
Fig. 5 and Lines 165-188: SAP chemistry has a significant impact on the mechanical characteristics. Do SAP particles with different sizes have similar chemistry in this study? Please mention SAP chemistry or mention the importance of this parameter. The strange trend observed in Fig. 5(b) can be explained by this parameter.
Page 4, Line 149: Please show some SEM images in the manuscript to compare the results of normal concrete with SAP concrete.
Page 7, Lines 217-219: Please revise the text. The experimental database presented in this study is not enough to use the “optimal” phrase. More experimental studies are necessary for future work to confirm the database. Please mention this in the “Conclusions” also.
Author Response

(The authors gave the same response as above.)

Round 2
Reviewer 1 Report
All questions answered
Author Response

(The authors gave the same response as above.)

Reviewer 2 Report
In the revised manuscript, authors made some necessary revision and basically addressed the responses to the questions my comments- even though I do not agree with the use of acrylics in these materials. Therefore, I fully accept the revised manuscript. Anyway, I wonder if a comment about the development of green environmental protection of concrete could be included.
My job as a reviewer is not to have the authors “do it my way” or to “impose my way of thinking about these matters”, but to make sure that what they expose is logical and scientifically founded, which it clearly is.
Author Response

(The authors gave the same response as above.)

Reviewer 4 Report
The authors appropriately improved the structure of the manuscript.
Author Response
Dear Editor and Reviewers
Thank you very much for your letter. The reviewers’ comments concerning our manuscript entitled “Effect of Superabsorbent Polymer on the Mechanical Performance and Microstructure of Concrete”. These comments are all valuable and very helpful for revising and improving our paper, as well as important guiding significance to our researches. We have studied comments carefully and have made corrections which we hope meet with approval. All changes are marked in the track changes version. Especially, this manuscript was edited for proper English language, grammar, punctuation, spelling, and overall style by one or more of the highly qualified native English speaking editors.
Best wished
Sincerely